# Extracellular Vesicular Transmission of miR-423-5p from HepG2 Cells Inhibits the Differentiation of Hepatic Stellate Cells

**DOI:** 10.3390/cells11101715

**Published:** 2022-05-23

**Authors:** Michal Safran, Rula Masoud, Maya Sultan, Irena Tachlytski, Chofit Chai Gadot, Ron Pery, Nora Balint-Lahat, Orit Pappo, Nahum Buzaglo, Ziv Ben-Ari

**Affiliations:** 1Liver Diseases Center, Chaim Sheba Medical Center Tel-Hashomer, Ramat-Gan 5262000, Israel; michal.safran@sheba.health.gov.il (M.S.); masoud.rula@sheba.health.gov.il (R.M.); mayasult@gmail.com (M.S.); irina.tachlytski@sheba.health.gov.il (I.T.); chofit.chai@mail.huji.ac.il (C.C.G.); nahum.buzaglo@sheba.health.gov.il (N.B.); 2Department of General Surgery, Chaim Sheba Medical Center Tel-Hashomer, Ramat-Gan 5262000, Israel; ron.pery@sheba.health.gov.il; 3Pathology Department, Chaim Sheba Medical Center Tel-Hashomer, Ramat-Gan 5262000, Israel; balint.lahat@sheba.health.gov.il (N.B.-L.); orit.pappo@sheba.health.gov.il (O.P.); 4Pathology Sackler School of Medicine, Tel Aviv University, Tel-Aviv 6329302, Israel

**Keywords:** liver fibrosis, extracellular vesicles, exosomes, miRNA-423-5p, hepatic stellate cells

## Abstract

Liver fibrosis (LF) is a major cause of morbidity and mortality worldwide. Hepatic stellate cells (HSCs) are the primary source of extracellular matrix in the liver and their activation is a central event in LF development. Extracellular vesicles (EVs) are intercellular communication agents, which play important roles in physiological processes in chronic liver diseases. The aim of this study was to examine the crosstalk between hepatocytes and HSCs mediated by hepatocyte-secreted EVs. EVs were purified from primary mouse hepatocytes, HepG2 cell lines, under normal or stressed conditions. The effect of EVs on primary HSCs (pHSCs) differentiation was evaluated by measuring of differentiation markers. In addition, their impact on the carbon tetrachloride (CCl4)-induced fibrosis mouse model was evaluated. The results demonstrated that HepG2-EVs regulate HSC differentiation and that under stress conditions, promoted pHSCs differentiation into the myofibroblast phenotype. The evaluation of miRNA sequences in the HepG2 secreted EVs demonstrated high levels of miR-423-5p. The examination of EV cargo following stress conditions identified a significant reduction of miR-423-5p in HepG2-EVs relative to HepG2-EVs under normal conditions. In addition, pHSCs transfected with miR-423-5p mimic and exhibit lower mRNA levels of alpha smooth muscle actin and Collagen type 1 alpha, and the mRNA expression level of genes targeted the family with sequence-similarity-3 (FAM3) and Monoacylglycerol lipase (Mgll). This study strengthened the hypothesis that EVs are involved in LF and that their cargo changes in stress conditions. In addition, miR-423-5p was shown to be involved in HSCs differentiation and hence, fibrosis development.

## 1. Introduction

Chronic liver diseases (CLDs), such as alcoholic liver disease, non-alcoholic fatty liver disease (NAFLD/NASH), and chronic viral hepatitis, can progress to the development of liver fibrosis (LF) [1,2]. LF is characterized by an excess production and deposition of extracellular matrix (ECM) proteins that derange the anatomical and vascular structure of the liver [2]. Studies have shown that LF can be a reversible process following the administration of an etiology-oriented treatment [3]. However, if left untreated, LF may lead to liver cirrhosis, organ failure, hepatocellular carcinoma (HCC), and death. According to the National Vital Statistics Report 2017 from the Center for Disease Control and Prevention in the USA, 1.8% of adults are diagnosed with CLDs and approximately 2 million deaths due to CLDs are reported worldwide each year [4].

All liver cell types including hepatocytes, fibroblasts, Kupffer cells, liver sinusoidal endothelial cells, biliary epithelial cells, and hepatic stellate cells (HSCs) have been demonstrated to be involved in LF, with HSCs being the most thoroughly studied and showing a strong association with LF [5]. HSCs are the main myofibroblast progenitor cells that comprise ~15% of liver cells [6]. In a healthy liver, HSCs are the primary storage site for retinoids and are present in a quiescent, non-proliferative state. However, following liver injury, HSCs become activated, take on a proliferative and contractile myofibroblast phenotype, and increase their synthesis of ECM proteins [6,7]. All chronic liver injury etiologies have been associated with HSC activation [8]. To date, the mechanism that underlies HSC transdifferentiation remains unclear. Exosomes are known as a mediator of cell–cell communication and are involved in cell migration, anti-viral infection, tumor growth, and regeneration via the horizontal transfer of their cargo [9,10]. Their cargo contains bioactive substances such as lipids, proteins, and microRNAs (miRNAs). In the liver, hepatocyte exosomes have shown to be involved in the regulation of cell proliferation, regeneration, and LF [11].

Recent evidence has demonstrated that miRNAs are involved in hepatocyte proliferation, differentiation, and lipid metabolism in HCC [12].

The present study focused on the relationships between hepatocytes and HSCs during fibrosis both in healthy and liver stress models. Using cell lines, primary HSCs, and mouse models, we confirmed the role of hepatic extracellular vesicles (EVs) in LF. Furthermore, we identified miR-423-5p in hepatic EVs and demonstrate its significant contribution to the regulation of LF.

## 2. Materials and Methods

### 2.1. Cell Culture

Human liver hepatocellular carcinoma cells (HepG2), human hepatic stellate cells (LX2), and human embryonic kidney (HEK) cells were maintained in Dulbecco’s Modified Eagle Medium (DMEM) (Biological Industries, Göttingen, Germany) containing 25 mM glucose, 4 mM l-glutamine, 10% Exosome-depleted FBS (System Biosciences, Palo Alto, CA, USA), 1% penicillin (Biological Industries, Göttingen, Germany), and 100 µg mL-1 streptomycin (Biological Industries). Cells were maintained at 37 °C in a humidified incubator with 5% CO_2_.

### 2.2. Extracellular Vesicle Isolation

Cell culture media from LX2, HepG2, primary mouse hepatocyte or primary mouse kidney cells (EV-depleted FBS (System Biosciences, Palo Alto, CA, USA)) were collected following a 5-day incubation. The media of each cell type were centrifuged at 2500× *g*, for 10 min, at 4 °C to remove debris. The collected supernatants were centrifuged again at 8050× *g* for 45 min, at 4 °C to remove cell debris and larger particles. The supernatant was then passed through a 0.45 μm syringe filter. The filtrate was concentrated using a VivaCell 100 kDa (Sartorius, Göttingen, Germany), at 2500 g, for 25 min, at 4 °C. The concentrated samples were collected into 13 mL Quick-Seal polyallomer tubes (Beckman Coulter, Irving, TX, USA) and the volume was brought to 13 mL with PBS. Next, samples were centrifuged at 100,000× *g*, at 4 °C, in a 50.2 Ti fixed-angle rotor (Beckman Coulter, Brea, CA, USA) overnight. The purified exosomes from each of the cell types were re-suspended in PBS, divided into aliquots and stored at −80 °C.

Extracellular vesicle secretion inhibition was performed as follows HepG2 and LX2 cells were maintained in cultured medium containing Exosome-Depleted FBS, (System Biosciences, Palo Alto, CA, USA) with 20 μM GW4869 in DMSO (Sigma-Aldrich, Rehovot, Israel) an inhibitor of exosome biogenesis/release (final DMSO concentration was 0.005%) [13]. Following a 5-day incubation at 37 °C, 5% CO_2_, EVs were isolated as described above.

Primed medium fractionation: The primed medium was collected following a 5-day incubation and separated into supernatant and pellet fractions or filtered through a cellulose membrane with a cutoff of 100 kDa as described above. pHSCs were incubated for 24 h with the supernatant, pellet, or the separated fractions containing >100 kDa or <100 kDa particles.

### 2.3. Nanosight

EV size distribution and concentration were measured with a NanoSight NS300 Instrument (Malvern Panalytical, Malvern, United Kingdom). The samples were diluted 1:100 and 1:1000 in sterile-filtered PBS and analyzed. The measurements were based on 5 duration videos of one minute long, screen gain 9.3, camera level 11, detection threshold 3, and screen gain 10. Data analysis was performed using the Nanoparticle Tracking Analysis (NTA) software.

### 2.4. Electron Microscopy

Negative-stained samples (HepG2 and LX2 EVs) were loaded on formvar/carbon-coated grids, fixed in 2% paraformaldehyde for 1 h in room temperature and washed with PBS. The sample then were contrasted with 2% aqueous uranyl acetate. Samples were examined under a JEM-1400 plus transmission electron microscope (TEM) (Jeol, Tokyo, Japan) [14].

### 2.5. Fluorescent Labeling

EVs were diluted in PBS (1:20) and gently mixed in the dark with 1 mg/mL thiazole orange (Sigma-Aldrich, Rehovot, Israel) in 100% ethanol (1:2000). EVs were then separated using an ultracentrifuge 50.2 Ti fixed-angle rotor at 100,000× *g*, 4 °C, 12 h. The pellet was suspended with cold PBS.

### 2.6. Primary Mouse Hepatic Stellate Cell Isolation

ICR mice (6–8 weeks old) (Envigo, Rehovot, Israel) were anesthetized with isoflurane (USP Terrell TM, Piramal, Lexington, KY, USA) and livers were dissected into Gey’s balanced salt solution buffer (GBSS) and washed three times. The liver samples were then incubated for 15 min at 37 °C in 100 mL GBSS200 (GBSS, supplemented with collagenase, pronase, and CaCl_2_ (Roche, Munich, Germany), 6000 U collagenase type 1 (Worthington Biochemical Corporation, Lakewood, NJ, USA), and 1 mM CaCl_2_ (MERCK, Rehovot, Israel). Following two washes, the samples were treated with 50 mL GBSS100 (GBSS, supplemented with collagenase (Enco), pronase (Dyn Diagnostics, Caesarea Industrial Park, Israel), DNase I (Sigma-Aldrich, Rehovot, Israel), and 0.5 mM CaCl_2_, for 30 min. Liver extracts were then filtered using a 40 μm mesh and centrifuged at 2000 rpm for 7 min, at room temperature (RT). Cell pellets were re-suspended in 7 mL OptiPrep^TM^ (Sigma-Aldrich, Rehovot, Israel) in GBSS buffer (1:5 ratio) and then centrifuged at 14,000× *g*, for 20 min, at RT. HSCs layer was transferred into a new tube, washed with 20 mL Dulbecco’s modified Eagle’s medium (DMEM), supplemented with 10% fetal bovine serum (FBS), 1% penicillin-streptomycin, and 1% glutamine (Biological Industries, Beit Haemek, Israel) and centrifuged for 7 min, at 2000 rpm, at RT. HSC pellets were then re-suspended in 10 mL culture medium (DMEM with 2% FBS, 1% penicillin-streptomycin, and 1% glutamine) and 4 ∗ 10 [6] cells/mL were cultured at 37 °C in a humidified incubator, with 5% CO_2_.

### 2.7. Primary Mouse Hepatocyte Isolation

ICR mice (6–8 weeks old) were perfused (5 mL/min) with 35 mL Hanks’ Balanced Salt solution (HBSS) (Biological-Industries, Beit Haemek, Israel) containing 2.1 g/l sodium bicarbonate and 0.2 g/L ethylenediaminetetraacetic acid (EDTA) for 7 min. Next, 50 mL liver digestive medium (Invitrogen, Waltham, MA, USA) was infused into the inferior vena cava. Livers were harvested and gently minced in plating medium (high-glucose DMEM, 10% FBS, 2 mM sodium pyruvate, 2% penicillin-streptomycin, 1μM dexamethasone (Sigma-Aldrich, Rehovot, Israel) and 0.001 μM insulin (Sigma-Aldrich, Rehovot, Israel)). Samples were passed through a 70 μm mesh filter and centrifuged at 0.4× *g*, for 5 min, at RT. The cell pellets were then re-suspended in 10 mL Percoll (Sigma-Aldrich, Rehovot, Israel), dissolved in PBS (1:10), and centrifuged again at 0.4× *g*, for 5 min, at RT. Hepatocytes were washed twice with 25 mL plating buffer, re-suspended in 10 mL plating medium, plated (4 ∗ 10 [5] cells/mL) on 6-well plates coated with collagen (Sigma-Aldrich, Rehovot, Israel), and cultured at 37 °C in a humidified incubator, with 5% CO_2_. Following 2 h of incubation, the medium was replaced with maintenance medium (high-glucose DMEM, 10% FBS, 0.2% bovine serum albumin (BSA), 2 mM sodium pyruvate, 2% penicillin-streptomycin, 1 μM dexamethasone and 1 nM insulin).

### 2.8. Primary Mouse Kidney Cell Isolation

ICR mice (8–12 weeks old) (Envigo, Rehovot, Israel) were anesthetized with isoflurane (USP Terrell TM, Piramal, Lexington, KY, USA), after which, kidneys were extracted and dissected into 25 mL digestion medium (71 mg collagenase type 4 (Worthington Biochemical Corporation, Lakewood, NJ, USA) per 50 mL DMEM) and incubated at 37 °C for 40 min. Next, the digested kidneys were filtered through a 70 μm mesh and centrifuged at 300× *g* for 5 min, at RT. Cell pellets were washed twice with DMEM and centrifuged again at 300× *g*, for 5 min, at RT. The cell pellets were then re-suspended in fresh medium and cultured in 6-well collagen-coated plates (Thermo Scientific™, Waltham, MA, USA).

### 2.9. Animal Experiments

All animal experiments were conducted in accordance with the ARRIVE guidelines and with the institutional guidelines for animal care. C57BL6 and ICR male mice (Envigo, Rehovot, Israel) were maintained in a pathogen-free facility and fed pellet food and water ad libitum.

### 2.10. Stress Conditions

#### 2.10.1. Treatment with ethanol (mimicking alcoholic steatohepatitis (ASH))

One day after seeding HepG2 cells in 6-well tissue culture plates, 80 mM EtOH was added to the medium for 24 h, after which, culture medium was replaced with fresh medium containing 80 mM EtOH and cells were cultured for another 24 h [15]. EVs were isolated following ethanol treatment as described previously (Section 2.2).

#### 2.10.2. Treatment with lauric acid (mimicking NASH)

One day after seeding HepG2 cells in 6-well tissue culture plates, the culture medium was replaced with culture medium containing 500μM lauric acid (Sigma-Aldrich, Rehovot, Israel) for 24 h [16]. EVs were isolated following ethanol treatment as described previously (see Section 2.1).

pHSCs were incubated for 24 h with EVs isolated from HepG2 treated with ethanol or lauric acid. RNA was isolated using Tri-reagent ((Sigma-Aldrich, Rehovot, Israel). cDNA of miR-423-5p was prepared using qScript^®^ microRNA cDNA Synthesis Kit (Quanatbio, Beverly, MA, USA) according to manufacturer’s instructions. The level of miR-423-5p was detected by RTq-PCR, with RNU6 as an internal control gene for normalization. Each of the described experiments was performed in biological triplicates and in technical triplicates.

### 2.11. Real-Time qPCR Analysis

Total RNA was extracted from liver specimens or from primary HSCs, using TRI-Reagent (Bio-Lab, Jerusalem, Israel). RNA concentration was determined using a NanoDrop spectrophotometer (Thermo Scientific™, Waltham, MA, USA, 2000/2000 c). cDNA was prepared by reverse transcription (qScript cDNA Synthesis Kit, Quantabio, Beverly, MA, USA) of 1 μg total RNA, according to the manufacturer’s instructions. qPCR was performed using the SYBR Green PCR Master Mix (Applied Biosystems, Waltham, MA, USA), according to the manufacturer’s recommendations, with specific primers for the αSMA, Col1a, TIMP1, TIMP2, PDGF, and TGFβ genes (listed in Table 1), in an ABI Step ONE Plus system (Applied Biosystems, Waltham, MA, USA). Relative mRNA quantification was performed using the ΔΔCT method (comparative ΔCT), with 18s as the internal control gene for normalization. Each experiment was performed in biological triplicates and in technical triplicates.

### 2.12. Western Blot Analysis

Cells or liver specimens were lysed with lysis buffer (RIPA—Sigma-Aldrich, Rehovot, Israel) containing a protease inhibitor cocktail and phosphatase inhibitors (Bimake, Houston, TX, USA). Protein levels were quantified using the QPRO-BCA kit standard (Cyanagen, Bologna, Italy). Proteins were separated by 12% SDS-PAGE, transferred to nitrocellulose membranes (Tamar, Mevaseret Zion, Israel), and then probed with monoclonal antibodies specific for αSMA (Sigma-Aldrich, Rehovot, Israel), HSC70 (Santa Cruz Biotechnology, INC., Santa Cruz, CA USA), or CD63 (BioVision, Inc., Milpitas, CA, USA), or a polyclonal anti-actin 1–19 antibody (Santa Cruz Biotechnology, INC., Santa Cruz, CA, USA) diluted 1:1000. Peroxidase-conjugated goat anti-rabbit or goat anti-mouse antibodies (1:10,000, Jackson Immunoresearch Laboratories, West Grove, PA, USA) were used as secondary antibodies. Immunoreactive bands were visualized using the Western blot chemiluminescence reagent (Cyanagen, Bologna, Italy). Relative quantification of the proteins was performed using the ImageJ software.

### 2.13. MicroRNA Transfection

pHSCs (4 ∗ 10 [6] cells/well in a 6-well plate) were plated 24 h prior to transfection. Interferin reagent (polyplus-transfection, Illkirch-Graffenstaden French) was used to transfect custom synthesized miRNA sequences (Ambion™, mirVana™, Fisher Scientific Cat# 4464066, MA, USA), as per the manufacturer’s instructions. miR-423-5p (5′-UGAGGGGCAGAGAGCGAGACUUU-3′ and a commercial negative control was used (AllStars negative control siRNA, cat# 1027280, Qiagen, Germantown, Maryland, USA).

Transfection efficiency: pHSCs membrane were labeled with PKH67 green fluorescent cell linker (Sigma-Aldrich, Rehovot, Israel) according to the manufacturer’s instructions. Following labeling, pHSCs were incubated with EVs containing labeled miRNA (Sigma-Aldrich, Rehovot, Israel) for 20 min. Next, fluorescent images of the labeled pHSCs and EVs were taken using confocal microscope (ZEISS, Jena, Germany).

### 2.14. Carbon Tetrachloride-Induced Liver Fibrosis Mouse Model

Eight-week-old male C57BL6 mice were administered intraperitoneal injections of 2 mL/kg carbon tetrachloride (Sigma-Aldrich, Rehovot, Israel), dissolved (1:4) in olive oil, twice a week for four weeks. Negative control mice were administered olive oil only. Each experimental group included 6–8 mice. Twenty-four hours following the eighth injection, 1.5 ∗ 10 [9] LX2 or HepG2-EVs dissolved in 200 μL PBS (Biological Industries, Beit Haemek, Israel) were injected into the tail vein. Negative controls were injected with 200 μL PBS. Mice were then treated with carbon tetrachloride twice a week for an additional two weeks. At the end of the experiment, blood samples were collected from the orbital sinus, mice were sacrificed, and the livers were isolated. Liver specimens were fixed in formalin for histological staining or snap-frozen in liquid nitrogen and stored at −80 °C for protein and RNA extraction. Serum aspartate transaminase (AST) and alanine transaminase (ALT) levels were determined using the Chemistry Analyzer AU5800 (BECKMAN COULTER), by standard procedures.

### 2.15. Masson’s Trichrome Staining

Paraffin-embedded liver tissues were sliced into 5 μm sections. Staining was performed using a commercial kit (Trichrome Stain (Masson) Kit, HT-15 (Sigma-Aldrich, Rehovot, Israel)). The slides were viewed with a Nikon TL microscope (×20), (Nikon Diaphot 200 Inverted Microscope, Natori, Japan) and photographed and the total fibrotic area in the tissue sections was measured with ImageJ software (NIH, Maryland, USA).

### 2.16. Statistical Analysis

Results are presented as the mean of at least three independent experiments. Two-tailed Student’s *t* tests were performed to analyze the statistical significance between groups, as appropriate. *p* < 0.05 was considered statistically significant. Error bars indicate standard error.

## 3. Results

### 3.1. Hepatocyte-Primed Medium Reduced Hepatic Stellate Cell Differentiation

To examine the effect of hepatocytes on HSC differentiation, primary HSCs (pHSCs) isolated from mice were incubated for 24 h with medium primed by primary hepatocytes, primary kidney cells, HepG2, LX2, or HEK. Thereafter, expression levels of the differentiation markers smooth muscle alpha-actin (αSMA) and collagen type a1 (Col1a) mRNA were measured using qRT-PCR. As can be seen in Figure 1A, αSMA and Col1a mRNA levels were significantly lower in pHSCs incubated with primary hepatocyte-primed medium in comparison to those incubated with medium primed by primary kidney cells. Similar results were obtained following exposure of pHSCs to HepG2-primed medium as compared to those incubated with HEK-primed medium (Figure 1B). In agreement with these results, αSMA protein levels were five-fold lower in pHSCs exposed to HepG2-primed medium as compared to those exposed to LX2-primed medium (Figure 1C). In addition, αSMA and Col1a mRNA levels were significantly lower in pHSCs exposed to HepG2-primed medium as compared to pHSCs incubated with LX2-primed medium (Figure 1D), suggesting that the crosstalk between hepatocytes and HSCs mediates the inhibition of HSC differentiation. Primary kidney cells and the HEK cell line were used as control groups. The selection of these cell types was made due to the fact that kidney cells are not present in the same microenvironment of liver cells. Therefore, it is likely that the effect of kidney EVs on liver cells is negligible and can be used as a group for the comparison of the effect of liver cells EVs on pHSCs.

### 3.2. Hepatocyte Extracellular Vesicles Mediate the Inhibition of Hepatic Stellate Cell Differentiation in Primed Medium

To test which component of the primed medium mediated the effect on pHSC differentiation markers, HepG2- and LX2-primed media were fractionated by ultracentrifugation or by 100 kDa molecular weight cut-off discs (see Section 2.2). Following the 24 h incubation of pHSCs with the HepG2- or LX2-primed media, supernatant fractions or pellet αSMA and Col1a mRNA levels were examined using qRT-PCR. αSMA and Col1a mRNA levels in pHSCs incubated with HepG2 or LX2 supernatant were similar (Figure 2A). However, upon incubation with the HepG2 pellet fractions, αSMA and Col1a mRNA levels were ~five folds lower relative to those seen in pHSCs incubated with LX2 pellets or with HepG2 supernatants (Figure 2A). In agreement with these observations, pHSCs exposed to HepG2-primed medium >100 kDa fraction exhibited ~3.5 folds lower αSMA and Col1a mRNA levels relative to those seen in pHSCs exposed to HepG2-primed medium (<100 kDa fraction). LX2-primed medium (>100 kDa or <100 kDa fraction) exhibited similar mRNA levels of αSMA and Col1a (Figure 2B).

To confirm that the effect of the pellet on pHSC fibrosis markers was mediated by EVs, LX2 and HepG2 cell lines were cultured in the presence of 20 μM GW4869, an established inhibitor of exosome biogenesis/release [13]. pHSCs incubated with EV-free LX2-primed medium showed αSMA mRNA levels similar to those measured in pHSCs incubated with LX2-primed medium containing EVs (Figure 3A). In contrast, αSMA mRNA levels in pHSCs incubated with EVs-free HepG2-primed medium were significantly higher relative to their levels in pHSCs incubated with HepG2-primed medium containing EVs (Figure 3B).

Nanoparticle tracking analysis (NTA) of EVs isolated from both HepG2 and LX2 pellets. The average HepG2 and LX2 EVs yield was 2.56 × 10^−9^ ± 1.50 × 10^−8^ particles for 1 L of medium. HepG2 and LX2 EVs measured a mean diameter of 95 ± 2.5 nm and an approximate size range of 50–300 nm and 50–420 nm for HepG2 and LX2, respectively (Figure 4A,B). Transmission electron microscopy (TEM) analyses confirmed these diameters and showed that most of the particles were round-shaped (Figure 4C,D). Western blot analysis confirmed the presence of the EV-associated protein CD63 in both pellets (Figure 4E).

Taken together, these results strengthen the hypothesis that EVs secreted by hepatocytes are the component that regulate HSCs differentiation.

### 3.3. Hepatocyte Extracellular Vesicles Inhibit the Development of Liver Fibrosis In-Vivo

To examine the effect of hepatocyte exosomes on LF in-vivo, C57BL6 mice exhibiting the carbon tetrachloride (CCl4) fibrosis were treated with CCl4, CCl4 + LX2-EVs, CCl4 + HepG2-EVs, or saline. Mice were injected with CCl4 (2 mL/kg) twice a week for 4 weeks. HepG2 and LX2 EVs (1.5 × 10^9^ particles) were injected once intravenously (IV). Following a one-time EV treatment, animals were treated with CCl4 for an additional 2 weeks in order to prevent the spontaneous regression of LF. At the end of the experiment, blood samples were collected, the animals were sacrificed, and the livers were isolated and subjected to RNA and protein purification and for histological staining. As expected, CCl4 significantly increased aspartate transaminase (AST) and alanine aminotransferase (ALT) protein levels in the mouse serum, the mRNA levels of the fibrosis markers αSMA, Col1a, PDGF, TGFβ, TIMP1, and TIMP2 in the mouse livers, and the αSMA protein levels in comparison to the control untreated group (Figure 5A–C). Parallel treatment with HepG2 EVs prevented the elevation of AST and ALT and the expression of the fibrosis markers in comparison to their level in the CCl4 group. It was also noticed that the HepG2-EVs treatment group exhibited similar levels of liver enzymes and mRNA level of fibrosis markers to those seen in the control group (untreated). In addition, the αSMA protein levels in the liver were examined. In agreement with αSMA mRNA level, the protein level in the HepG2-EVs treated group was significantly lower than in the CCl4 toxin model group and was similar to those seen in the control group. In contrast, LX2-EVs did not prevent the elevation of AST, ALT, the expression of the fibrosis markers, and the increase in αSMA protein levels, which proved similar to those measured in mice treated with CCl4 only (Figure 5A–C).

Masson’s trichrome staining of mouse liver sections showed that the relative area of fibrotic tissue following CCl4 treatment was three times higher than the relative area in the untreated control group (Figure 6B). In contrast, HepG2-EV treatment resulted in a significant reduction in the relative fibrotic area in comparison to mice treated with CCl4 only and were similar to the area measured in untreated controls. Moreover, mice treated with LX2-EVs exhibited LF areas similar to those in the group treated with CCl4 only. Taken together, these results demonstrate that HepG2-EVs are involved in the regulation of LF in-vivo.

EVs play a central role in physiological and pathophysiological processes. EV components as proteins and miRNAs transferred to target cells regulate physiological events, such as intracellular signaling, cell proliferation, and differentiation [10]. Here, we examined the effect of miRNAs isolated from both HepG2 and LX2 EVs on pHSCs differentiation markers. miR sequences purified from HepG2 and LX2 EVs and transfected into pHSCs. HepG2-EVs purified miRNA sequences significantly reduced αSMA mRNA levels in pHSCs as compared to LX2 EV-purified miRNA (Figure 7A).

Sequencing of EV miRNAs from LX2-EVs and HepG2-EVs identified 103 miRNAs that were present at higher concentrations in HepG2-EVs as compared to LX2-EVs (data not shown). Following miRNA target prediction analysis, miR-423-5p was selected for further examination. According to the miRBase (the microRNA database), one of the predicted targets for miR-423-5p is Cola1α (Target rank 36). The miR-423-5p expression levels in HepG2-EVs compared to LX2-Evs, as determined using qRT-PCR, were 3.8-fold higher in the HepG2-Evs (Figure 7B). In addition, pHSCs transfected with a miR-423-5p mimic showed significantly lower αSMA, Col1a, Mgll, and FAM3 mRNA levels compared to untransfected pHSCs (Figure 7C). These observations may imply that the distinct sets of miRNA in the various liver cell subtypes, are involved in the differentiation of pHSC.

### 3.4. Stress Reduces miRNA-423-5p Levels in HepG2-EVs

Previously published studies have demonstrated the role of EVs released by stressed hepatocytes (NASH (non-alcoholic steatohepatitis) and ASH (alcoholic steatohepatitis) mimicking conditions) in HSC transdifferentiation [17]. To evaluate whether stress conditions affect miR-423-5p levels in HepG2-EVs. HepG2 cells were exposed to either 80 mM EtOH or 500 μM lauric acid for 5 days. Under both stress conditions, the levels of EV-isolated miR-423-5p were significantly lower in comparison to their levels in unstressed HepG2-EVs (Figure 8A). pHSCs that were exposed to stressed HepG2-EVs showed significantly higher αSMA and Col1a mRNA levels in comparison to EVs secreted from unstressed HepG2. Similar trends were noted with regards to the mRNA levels of Mgll and FAM3 genes, which are reported targets of miR-423-5p and implicated in LF [18,19,20]. These results demonstrate again that miR-423-5p is involved in LF development.

## 4. Discussion

LF, and particularly cirrhosis, is a widespread pathology that has an impact on health in patients with CLDs. Its prevention or reversal has been the focus of the development of specific drugs for this patient population [21]. Despite advances in therapy, morbidity and mortality remain high [22,23]. Although it is well known that LF, including advanced stages, is a reversible process, little is known about the mechanism that maintains HSCs in a quiescent state, or that allows HSCs to revert back to this state following the resolution of liver stress. As mentioned above, EVs are significant players in CLDs and are widely studied for their role in liver diseases [10].

The present study showed that HepG2-secreted EVs regulate HSC differentiation and thus affect LF. This was demonstrated both in-vitro using mouse liver-derived pHSCs and in-vivo using the mouse CCl4-induced LF model. In addition, we identified high levels of miR-423-5p in HepG2-EVs, whose levels were reduced following stress conditions. Moreover, **α**SMA, Col1a, FAM3, and Mgll expression were shown to be regulated by miR-423-5p.

Due to the central role of EVs in intercellular communication and their involvement in major cellular events, the potential of EVs, and particularly of exosomes, to serve as biomarkers or therapeutics has grown exponentially over the past decade [24]. Recent evidence has shown that EVs are also produced by several liver cell types including hepatocytes, HSC, liver sinusoidal endothelial cells, and cholangiocytes [25]. Hepatocyte exosomes were reported to mediate therapeutic changes in activated HSCs or injured hepatocytes that occur downstream of heparin- or integrin-dependent binding interactions [22]. Hiroyuki et al. reported that hepatocyte exosomes deliver the protein synthetic machinery to form sphingosine-1-phosphate in target hepatocytes, resulting in cell proliferation and liver regeneration after ischemia/reperfusion injury or partial hepatectomy [23]_._ Our results are consistent with previous findings of the involvement of HepG2-EVs in the reduction of HSC differentiation marker levels and hence, in LF [10]. (Figure 2, Figure 5 and Figure 6). These findings imply that the maintenance of HSCs in a quiescent state is an active process regulated by hepatocyte EVs in the healthy liver. Recently it has been suggested that quiescent HSCs produce exosomes that inhibit the activation of HSCs and attenuate intracellular pathways of fibrogenesis [26,27]. In our study, LX2-EVs did not reduce fibrosis markers or LF (Figure 2, Figure 5 and Figure 6). These results may be due to species differences between the human LX2 cell line and mouse pHSCs. As mentioned above, EV cargo contains biological molecules such as lipids, proteins, and RNA, including mRNA and microRNA [9], each of which may mediate HSC differentiation [28]. In addition, it has been demonstrated that the secretion pattern and the cargo components of hepatocyte EVs changed following stressful events, such as hepatitis C virus infection, ASH, and NAFLD and were associated with HSC activation [29,30]. Our results demonstrate that EV-isolated RNAs are significant players in the regulation of pHSC differentiation (Figure 7). In agreement with our results regarding HepG2-EVs, the effect of HepG2 EV-derived RNA on HSC differentiation was significantly higher relative to LX2-EVs RNA. Further examination of the miRNAs in the EVs cargo, under optimal or following stress conditions, demonstrated that LX2-EVs and HepG2-EVs express different sets of miRNAs (data not shown), strengthening the hypothesis that the EVs cargo is unique to each cell type and phenotype. Since our experiments were performed with transformed tumoral cell lines, further examination of primary hypothesis EV cargo is needed. miRNAs are best known for eliciting gene silencing by lowering the stability and/or translation of mRNA, which they partially complement [31]. Few studies have demonstrated the role of miRNAs in the inhibition of HSC transdifferentiation. In 2016, Hyun et al. demonstrated the involvement of miRNA-378 in the inhibition of HSC activation and LF, as was measured by a decline in their levels in CCl4 fibrosis mouse model. Furthermore, they showed that miR-378a-3p suppressed HSC de-differentiation by targeting Gli3 expression [32]. The same year, Roy et al. reported on the downregulation of both miR-30C and miR-193 in experimental hepatofibrogenesis and in human LF. Their work identified transforming growth factor (TGF-β2) and Snail Family Transcriptional Repressor 1 (SNAIL1) as potential target genes of these miRNAs [33]. Our findings are in alignment with these reports. However, in this study, we demonstrated that miR-423-5p is present in higher levels in HepG2-EVs relative to LX2-EVs (Figure 7B). Currently, little is known about the physiological or pathophysiological role of miR-423-5p in the liver and in LF. Therefore, in this study, based on the effect of HepG2-primed medium and HepG2-EV on pHSC on differentiation markers in-vivo and in-vitro and the relative high level of miR-423-5p in HepG2-EVs, we chose to further examine the involvement of miR-423-5p in LF. miR-423-5p exhibited an inhibitory effect on osteosarcoma and ovarian cancer proliferation [34,35,36] and was demonstrated to regulate FAM3A-ATP-Akt signaling in cultured hepatocytes [18]. In addition, miR-423-5p levels were upregulated during HSCs activation and were downregulated following sulforaphane administration [37].

The present study expanded the understanding of gene targeting by miR-423-5p in the liver. We showed that Col1a, **α**SMA, Mgll, and FAM3 mRNA levels were significantly lower in pHSCs following transfection with miR-423-5p (Figure 7C). Mgll and FAM3 are known for their involvement in LF. The inhibition of Mgll was demonstrated to slow fibrosis progression and to reverse established LF [20]. In addition, in the livers of obese mice and steatotic patients, miR-423-5p expression was increased with decreases in FAM3A expression [18]. Furthermore, our findings demonstrated that stress conditions led to a reduction in miR-423-5p levels in HepG2-EVs and as a result, to an elevation of Col1a, **α**SMA, Mgll, and FAM3 mRNA levels in pHSCs (Figure 8). Taken together, our results suggest that miR-423-5p is involved in the regulation of LF and in maintaining HSCs in a quiescent state. Further study of the downstream signaling pathways in HSC transdifferentiation and the upstream factors involved in the synthesis of miR-423-5p in the hepatocytes will be essential to gain a better understanding of miR-423-5p mechanism of action in the regulation of HSCs differentiation.

## Figures and Tables

**Figure 1 cells-11-01715-f001:**
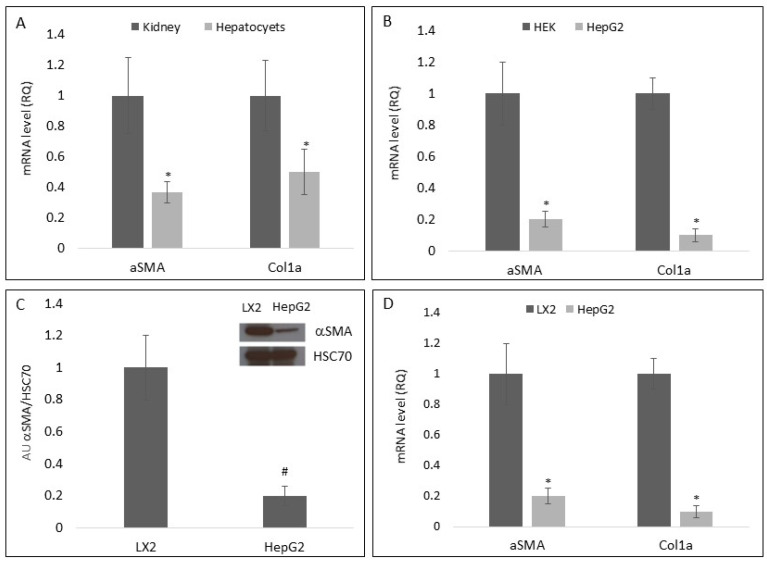
Effect of primed medium on pHSCs differentiations markers. Primary hepatocytes and HepG2 and LX2 cell lines were cultured for 5 days. The primed medium was collected and placed over pHSCs for 24 h. Primed medium from primary mice kidney cell cultures or HEK cell line were used as a control group. Following incubation, pHSCs were harvested and the mRNA or protein levels of αSMA and Col1a were examined. (**A**) The effect of hepatocyte-primed medium on αSMA and Col1a mRNA levels in pHSCs. (**B**) The effect of HepG2 cell-primed medium on αSMA and Col1a mRNA levels in pHSCs. (**C**) The effect of HepG2 and LX2 cell-primed medium on αSMA protein levels in pHSCs. (**D**) The effect of HepG2 and LX2 cell-primed medium on αSMA and Col1a mRNA levels in pHSCs. * Significantly lower than the kidney, HEK or LX2 cells *p* < 0.0001; ^#^ Significantly lower than LX2 cells, *p* < 0.008. ANOVA with a post-hoc Bonferroni-adjusted Student’s *t*-test.

**Figure 2 cells-11-01715-f002:**
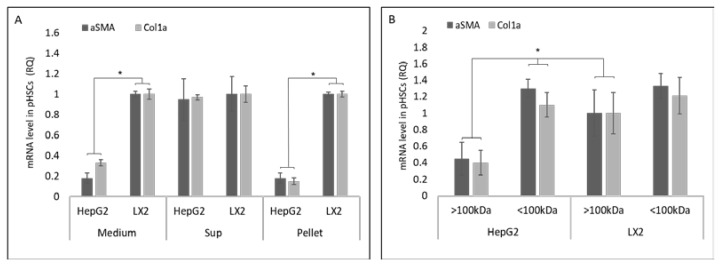
Effect of LX2 and HepG2-primed medium fractions on HSC differentiations markers. HepG2 and LX2 cells were cultured for 5 days. The primed medium was collected and separated into supernatant (Sup) pellet fractions or filtered through a cellulose membrane with a cutoff of 100 kDa (Section 2.2). pHSCs were incubated for 24 h with the supernatant, pellet, or the separated fractions containing >100 kDa or <100 kDa particles. (**A**) The effect of fractions of primed medium on αSMA and Col1a mRNA levels in pHSCs. (**B**) The effect of primed medium containing >100 kDa or <100 kDa particles on αSMA and Col1a mRNA levels in pHSCs. * Significantly lower than HepG2 cells or HepG2 >100 kDa, *p* < 0.05. ANOVA with a post-hoc Bonferroni-adjusted Student’s *t*-test.

**Figure 3 cells-11-01715-f003:**
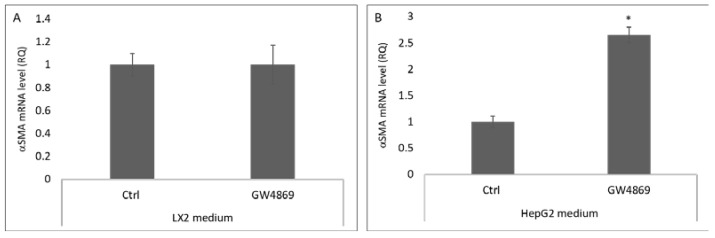
Effect of LX2 and HepG2 Extracellular Vesicles on αSMA in pHSCs. HepG2 and LX2 cells were incubated for 5 days with 20 μM GW4869, an inhibitor of EV secretion. pHSCs were then incubated with HepG2- or LX2-primed medium for 24 h and harvested, after which, αSMA mRNA levels were examined. (**A**) The effect of EV-free LX2-primed medium on αSMA mRNA levels in pHSCs (**B**). The effect of EV-free HepG2-primed medium on αSMA mRNA levels in pHSCs. * Significantly higher than the control, *p* < 0.01. ANOVA with a post-hoc Bonferroni-adjusted Student’s *t*-test. 2.5.

**Figure 4 cells-11-01715-f004:**
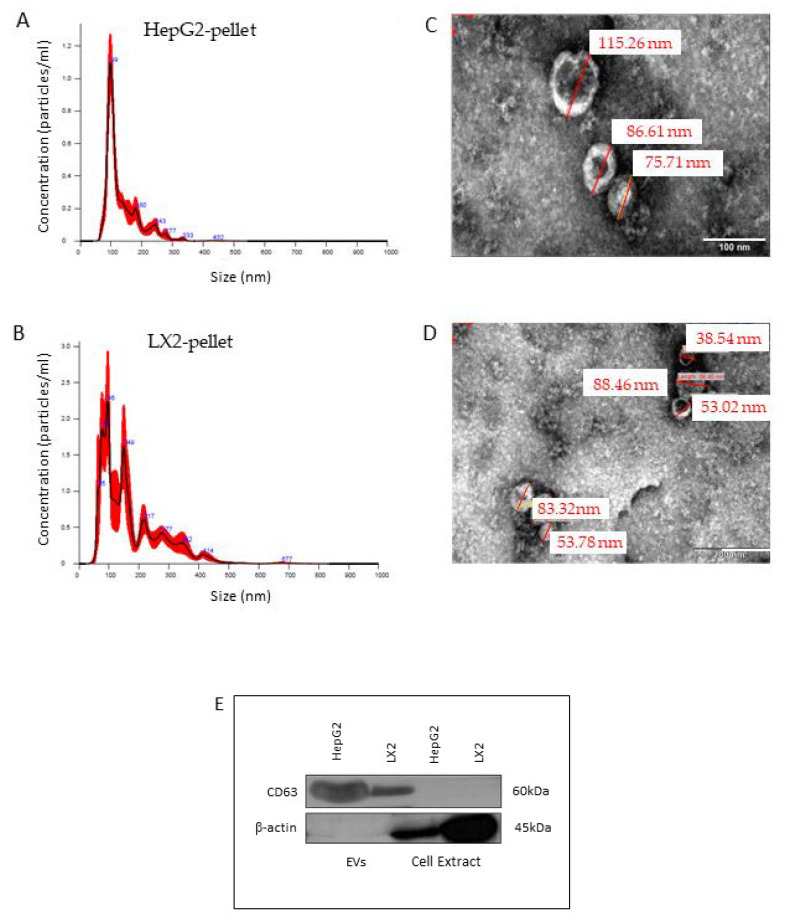
Characterization of LX2- and HepG2-primed medium pellets. Representative nanoparticle tracking analysis (NTA) of size distribution of particles isolated from HepG2- (**A**) and LX2- (**B**) primed medium. Representative transmission electron microscopy image of EVs from HepG2 (**C**) and LX2 (**D**) cultures. Bar = 100 nm. (**E**) The expression of CD63 in the isolated particles was determined by western blotting.

**Figure 5 cells-11-01715-f005:**
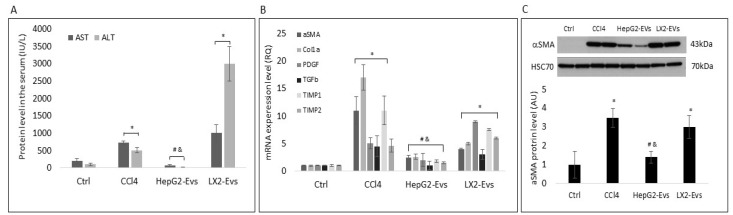
The effect of LX2 and HepG2 Extracellular Vesicles on liver fibrosis markers in-vivo. Eight-week-old male C57BL6 Mice were injected with 2 mL/kg CCl4 twice a week for 4 weeks. After the last CCl4 injection, EVs from LX2 and HepG2 cell lines (1.5 × 10^9^ particles) were IV administered to mice. CCl4 was injected for 2 additional weeks. At the end of the experiment, mice were sacrificed, livers were isolated, and mRNA and protein were extracted. (**A**) The effect of EVs on liver enzyme in CCl4 toxin model. (**B**) The effect of EVs on HSC differentiation markers in CCl4 toxin-induced fibrosis model. (**C**) The effect of EVs on αSMA protein level in CCl4 toxin-induced fibrosis model. ^#^ Significantly lower than CCl4, *p* < 0.05; ^&^ Significantly lower than LX2-Evs, *p* < 0.05; * Significantly higher than Ctrl, *p* < 0.05 (ANOVA with a post-hoc Bonferroni-adjusted Student’s *t*-test).

**Figure 6 cells-11-01715-f006:**
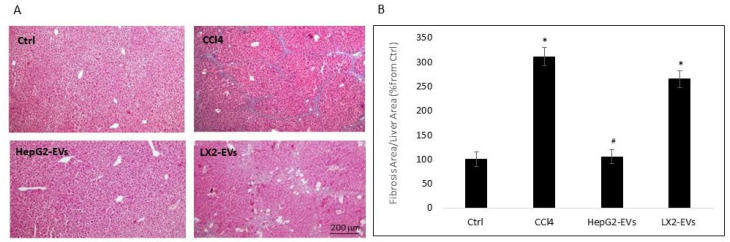
The effect of LX2 and HepG2 EVs on liver fibrosis in-vivo. (**A**) Representative Masson’s trichrome staining of mouse liver tissue following EV treatment described in Figure 5 legend. Using ImageJ software liver images were analyzed and the relative area of the fibrosis were measured. (**B**) Quantitative analysis of the liver fibrosis area, calculated as mean percent fibrotic area relative to that of the sham group. Results are expressed as mean ± SEM. * Significantly higher than sham; ^#^ Significantly lower than CCl4, *p* < 0.05 (ANOVA with a post-hoc Bonferroni-adjusted Student’s *t*-test).

**Figure 7 cells-11-01715-f007:**
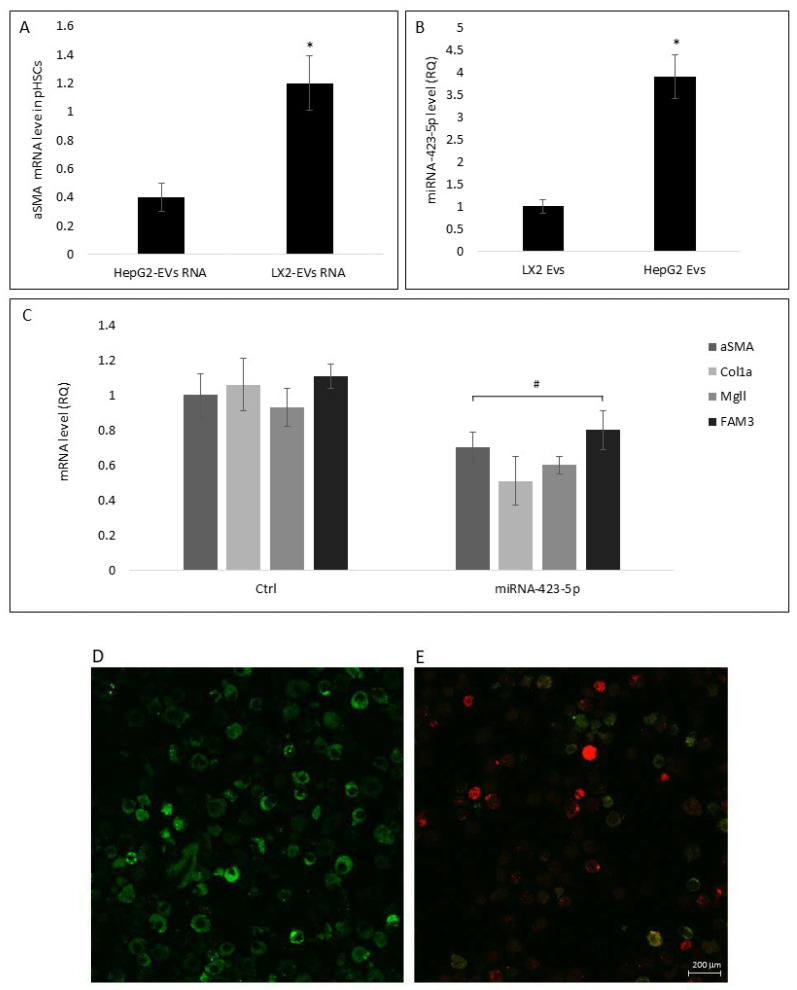
The effect of miRNA purified from HepG2-EVs, LX2-EVs, or miR-423-5p mimic on pHSC differentiation markers. miRNA sequences purified from HepG2-EVs or LX2-EVs were transfected into pHSCs. Following transfection (24 h), pHSCs were harvested and αSMA mRNA levels were determined. (**A**) The effect of miRNA purified from HepG2-EVs and LX2-EVs on αSMA mRNA levels in pHSCs. (**B**) miR-423-5p levels in LX2 and HepG2 EVs. RNU6 was used as a reference gene. (**C**) pHSCs were transfected with either 20 pM miR-423-5p mimic or non-relevant miRNA sequences as a control)**.** Following transfection (24 h), pHSCs were harvested and αSMA, Col1a, Mgll, and FAM3 mRNA levels were determined using RT-PCR. (**D**). Labeled pHSCs (green) (**E**). Merge image of labeled pHSCs (green) with labeled microRNA (red). Results are expressed as mean ± SEM of three independent experiments. * Significantly lower than LX2-EVs; ^#^ Significantly lower than Ctrl, *p* < 0.05 (ANOVA with a post-hoc Bonferroni-adjusted Student’s *t*-test).

**Figure 8 cells-11-01715-f008:**
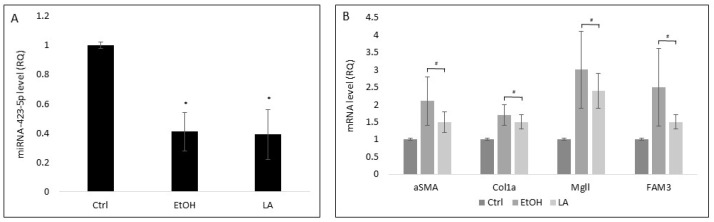
The effect of stress model on miR-423-5p levels and differentiation markers in pHSCs. HepG2 cells were exposed to either 80 mM EtOH or 500 μM lauric acid for 5 days, after which, HepG2 EVs were purified. HepG2 EVs miRNA was purified and quantified and miR-423-5p levels were determined. (**A**) The effect of stress on miR-423-5p levels in HepG2 EVs. (**B**) pHSCs were incubated (24 h) with HepG2 EVs purified following HepG2 exposure to stress conditions. pHSCs were then harvested and αSMA, Col1a, Mgll, and FAM3 mRNA levels were quantified. Results are expressed as mean ± SEM of 3 independent experiments. * Significantly lower than Ctrl; ^#^ Significantly higher than Ctrl, *p* < 0.05 (ANOVA with a post-hoc Bonferroni-adjusted Student’s *t*-test).

**Table 1 cells-11-01715-t001:** Primers used for quantitative real-time qPCR.

Gene	Reverse	Forward
18s	CAATCCAATCGGTAGTAGCG	GTAACCCGTTGAACCCCATT
αSMA	GTCAGGCAGTTCGTAGCTCTTCT	CTACTGCCGAGCGTGAGATTG
Col1a	GAGGCACAGACGGCTGAGTAG	CTGACTGGAAGAGCGGAGAGTAC
TIMP1	TGGTATCTGCTCTGGTGTGTCTCT	TGATTTCCCCGCCAACTC
TIMP2	TTCTGCCTTTCCTGCAATTAGATACT	CACGGCCCCCTCTTCAG
PDGF	GACTCATAATCTTCAGCTCGGACAT	CACCATGAAAGTGGCTGTCAA
TGFβ	ACCTTTGCCAATGCTTTCTTGTA	TCACTAGATCGCCCTTTCATTTC
FAM3A	GATACAGCCTTCCATCTCCAG	GATCTAGCCTTCCGTGACAG
Mgll	GTCAACCTCCGACTTGTTCC	TGATTTCACCTCTGGTCCTTG

## Data Availability

Not applicable.

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
