# Peer review of "Extracellular Vesicular Transmission of miR-423-5p from HepG2 Cells Inhibits the Differentiation of Hepatic Stellate Cells"

_cells, 2022, doi:10.3390/cells11101715_

Round 1

Reviewer 1 Report

The authors examined the crosstalk between hepatocytes and HSCs mediated by hepatocyte-secreted EVs.More details are needed.

major points

1. The miRNA 433 target prediction analysis is not shown in the manuscript.

2. How did the authors  detect level of miRNA-423-5 ?

3. Did  miRNA-423 directly reduce the αSMA, Col1a, Mgll and FAM3 mRNA ? More evidence is needed.

minor points

  1. Figure C&D are not clear.
  2. Line 234 RPq-PCR ?

Author Response

Thank you so much for the comments for the article "Extracellular Vesicular Transmission of miRNA-423-5p from HepG2 Cells Stimulates De-differentiation of Hepatic Stellate Cells" which was submitted for publication in "cells".

  1. The miRNA 433 target prediction analysis is not shown in the manuscript.
    For miR-423-5p target prediction analysis we use "miRBase" the microRNA database. We found 700 predicted targets for hsa-miR-423-5p in this database (http://mirdb.org/cgibin/search.cgi?searchType=miRNA&full=mirbase&searchBox=MIMAT0004748). Among the 700 genes, col1a was the only predicted target gene. However, we did not test the direct effect of miR-423-5p on its target genes. In addition, since no broad examination of miR-423-5p targets genes were performed, we chose to focus on HSC de-differentiation markers.
  2. How did the authors detect level of miRNA-423-5?
    RT-PCR. See new section 2.10
  3. Did miRNA-423 directly reduce the αSMA, Col1a, Mgll and FAM3 mRNA ? More evidence is needed.
    We did not test the mechanism of miR-423-5p on the tested genes. We assume that the only direct effect of miR-423-5p is on col1a gene. Understanding of the complete pathway of miR-423-5p is one of our future plans.

Reviewer 2 Report

In the manuscript ID cells-1650181 entitled “Extracellular Vesicular Transmission of miRNA-423-5p from HepG2 Cells Stimulates De-differentiation of Hepatic Stellate Cells”, the authors reported the importance of miRNA in EVs from hepatocyte on the hepatic stellate cells. Some of the observations are interesting, however, there are some concerns about the experiments:

Major points

In Fig1, the expressional level of Col1 and alpahSMA was decreased. Was there any morphological change in Hepatic stellate cells?

What was the reason for the increase in AST when LX2 medium was added in Fig. 5A?

In Fig6, EV from HepG2 shows improvement in fibrosis. Is only the expression of miRNA423-5p reduced in stressed HepG2? How about other miRNAs? Or is the total number of EVs reduced?      

In Fig. 8, the expression of miRNA423-5p is decreased from hepG2 in the stressed condition, how was this standardized?

Is the same amount of EV administered in Fig8B? Or is the amount of medium the same?

Do the EVs produced by hepatocytes in vivo and the miRNAs 423-5p contained in them have any effect on cells other than the liver, especially collagen-producing cells in other organ?

Minor points.

It is believed that clearer results will be seen when the effect of Inhibitor on miRNA423-5p is evaluated. It is not necessary to add this in this manuscript, but it might be effective to consider it in new studies in the future.

Line465 , there is a misuse of font, “alpha” SMA.

Line236 missing line.

Author Response

Thank you so much for the comments on the article "Extracellular Vesicular Transmission of miR-423-5p from HepG2 Cells Stimulates De-differentiation of Hepatic Stellate Cells".

In Fig1, the expressional level of Col1 and alpahSMA was decreased. Was there any morphological change in Hepatic stellate cells?
Unfortunately, at the present work we did not test morphological changes. These days we examine additional effects of primed medium and miRNA423-5p on HSC.

What was the reason for the increase in AST when LX2 medium was added in Fig. 5A?
According to our results, LX2 primed medium or LX2-EVs did not affect differentiation markers in-vitro (Fig1), in-vivo (Fig5B) or liver fibrosis (Fig6). In addition, LX2 did not worsened the tested liver indices (accept AST level). Hence, we assumed that LX2 did not contribute to an additional liver damage.  In addition, AST elevation may result following muscle injury. Since we did not test additional factors, we cannot attribute this elevation to other injury.  

In Fig6, EV from HepG2 shows improvement in fibrosis. Is only the expression of miRNA423-5p reduced in stressed HepG2? How about other miRNAs? Or is the total number of EVs reduced? 
We did not test other microRNA in HepG2-EVs following stress conditions.      

In Fig. 8, the expression of miRNA423-5p is decreased from hepG2 in the stressed condition, how was this standardized?

Is the same amount of EV administered in Fig8B? Or is the amount of medium the same?
At this experiment we used same amount of primed medium assuming that the amount of EVs were similar. 

Do the EVs produced by hepatocytes in vivo and the miRNAs 423-5p contained in them have any effect on cells other than the liver, especially collagen-producing cells in other organ?
At the present work we did not test the effect of miR-423-5p on other organ. Accordion the the available data on the literature miR-423-5p involved in several of cellular process. In addition, according to "miRBase" the microRNA database collagen type IV alpha 6 chain is one of miR-423-5p target gene. In our In future work we will test its effect in other organs.

Reviewer 3 Report

This paper aims to describe the effect of extracellular vesicles (exoxomes) derived from HepG2 and LX2 cells in the differentiation phenotype of primary hepatic stellate cells (HSC). Moreover, in the characterization of the components of these vesicles, authors find that those derived from HepG2 cells contain a significant increase in the levels of miRNA-423-5p, which are involved in the regulation of the expression of some profibrotic related genes from the HSC cells.

The results described in the paper would be relevant for scientist working in liver disease, more precisely in liver fibrosis.

Overall, this manuscript is well written, the work has been well designed, the data are original, the methodology employed is adequate, and the results are properly analyzed and described. In my opinion, this work could be acceptable for Cells when the following issues are adressed.

As mentioned, there are some concepts to be explained (perhaps in the Discussion section)

  • Authors have employed the results obtained from EV from kidney cells (primary cells or HEK) as the “Control cells” to compare the results obtained from EV obtained with the “hepatocytes cells”. The results obtained from EV from kidney cells might not be respresentative to be considered as Control, as hepatic stellate cells does not physiologically receive any inputs from kidney cells. The EV that are exported by kidney cells contain relevant information to neighboring cells, that could affect the normal biology of HSC cells, because is not intendent to reach these cell type. Please, discuss the use of this Control.
  • All along the manuscript the results obtained from EV coming from HepG2 cells are “assimilated” to “hepatocytes”. However, this cell type is a transformed tumoral cell line. Authors mention in line 552 that “EVs cargo is unique to each cell type and phenotype”. Then, EV coming from HepG2 cells could not be representative to the content that could be obtained from non-tumoral quiescent hepatocytes. Please, discuss the use of a tumoral cell line as Control.
  • It would be interesting to know if there is an effect associated to the sex of the animals (as only males have been employed in the work). This point can be introduced in the Discussion section.
  • It would be interesting to know what would have happened if authors would have compared the effects on HSC cells from EV coming from normal hepatocytes and EV coming from CCL4 rat treated hepatocytes.

Moreover, there are some minor points to be completed:

Material and Methods section:

  • There is a mixture of “rpm” and “g” values in the description of the centrifugation techniques. Please, provide the “g” value in all the centrifugation procedures.
  • Page 2 Section 2.1: The yield of the process of obtention of the EVs would be appreciated. Thus, the initial number of cells/type flasks employed and the total amount (or concentration) of EVs obtained in the procedure (for each cell type) would be of the interest of readers that could be interested in replicating some results.
  • Page 2 Section 2.5. Line 95. 20 µM. The symbol µ is lost
  • Page 4 Section 2.10. Line 153. Please, provide the source of the different cell lines employed in the work.
  • Page 5 Section 2.13. Please, provide the specific references for all the antibodies employed in the work. At least, the HSC70 and CD63 are missing.
  • Page 5 Section 2.16. 5 µm. The symbol µ is lost
  • Page 5 Section 3.1 of the results. Please, include in the Material and Methods section the concept of “medium primed” employed in the Results section.

Results section

  • Please, provide the exact “n” value for Figures 1, 2, 3, 4, 5 and 6.
  • Page 5 Section 3.1. Lines 235-236: There is a problem with the format of the sentence.
  • Page 6 Figure 1.Line 273. The figure footnote is wrong. Figure 1C corresponds to a Western Blot (protein levels), and the text appears to be “mRNA levels”
  • Page 6 Figure 1.Line 275-6. “# Significantly lower than the primary kidney cell or HEK cells”. This description does not correspond to the comparison in the Figure. In fact, # compares LX2 cells to HepG2 cells.
  • Page 7 Figure 2.The two first columns of Figure 2 are the same that those from Figure 1D?
  • Page 7 Figure 2. “*Significantly lower than the than the primary kidney cell or HEK cells.” These cells types are not employed in the experiment. Please, revise.
  • Page 7 Section 3.2. SMA The symbol α is lost
  • Page 7 Section3.2. Line 329. The inhibitor GW4869. Is it water soluble? Have authors employed any solvent (and added to control cells)?
  • Page 8 Figure 4. The text from the axes is not readable
  • Page 8 Figure 4b. “LX2-pelle” should be “LX2-pellet”
  • Page 8 Section 3.3. Line 379. “Inhibited” should be “prevented”
  • Page 8 Section 3.3. Line 383. In addition, the α SMA protein level in the liver were examined. This Figure is not present in the original version of the manuscript.
  • Page 9 Figure 6. The images does not present the bar size and the statistical symbols are not clear.
  • Page 10 Section 3.3. Lines 428-430. “miRNA molecules purified from HepG2 and LX2 EVs and transfected into pHSCs (Figure 6) significantly reduced α SMA mRNA levels as compared to LX2 EV-purified mRNA (Fig-7A)”. Please, rephrase this sentence.
  • Page 10 Figure 7. Lines 465 and 471. The symbol α is lost
  • Page 11 Figure 8. Lines 502 and 506. The symbols α and µ are lost

Discussion section

  • Page 12. Line 532. The word “synthetic” presents a different format/style
  • Page 12. Line 567. The word “sigling” should be “signaling”
  • “Supplementary Materials: The following supporting information can be downloaded at: www.mdpi.com/xxx/s1, Figure S1: title; Table S1: title; Video S1: title”

This referee have not seen the supplementary material.

Author Response

Thank you so much for the comments for the article "Extracellular Vesicular Transmission of miRNA-423-5p from HepG2 Cells Stimulates De-differentiation of Hepatic Stellate Cells" which was submitted for publication in "cells".

  • Authors have employed the results obtained from EV from kidney cells (primary cells or HEK) as the “Control cells” to compare the results obtained from EV obtained with the “hepatocytes cells”. The results obtained from EV from kidney cells might not be respresentative to be considered as Control, as hepatic stellate cells does not physiologically receive any inputs from kidney cells. The EV that are exported by kidney cells contain relevant information to neighboring cells, that could affect the normal biology of HSC cells, because is not intendent to reach these cell type. Please, discuss the use of this Control.
    As you mention, we chose kidney cells and HEK as cells that does not physiologically directly sent inputs to liver cells. The result from the experiments with these cells at the beginning of our work strength the hypothesis that communication between liver cells is a significant mechanism in liver fibrosis. We added note in section 3.1.  

  • All along the manuscript the results obtained from EV coming from HepG2 cells are “assimilated” to “hepatocytes”. However, this cell type is a transformed tumoral cell line. Authors mention in line 552 that “EVs cargo is unique to each cell type and phenotype”. Then, EV coming from HepG2 cells could not be representative to the content that could be obtained from non-tumoral quiescent hepatocytes. Please, discuss the use of a tumoral cell line as Control.

We added the following sentence: "Since that our experiment were performed with transformed tumoral cell lines, further examination of primary hypothesis EVs cargo is needed"

  • It would be interesting to know if there is an effect associated to the sex of the animals (as only males have been employed in the work). This point can be introduced in the Discussion section.

Thank you for this comment! According to the fact that cirrhosis is less frequent in women than in men it will be much interesting to make a comparison between genders in terms of microRNA changes.

  • It would be interesting to know what would have happened if authors would have compared the effects on HSC cells from EV coming from normal hepatocytes and EV coming from CCL4 rat treated hepatocytes.

These days we are preparing to conduct such an experiment. Such an experiment will allow a better understanding of the phenomenon described.

  • Page 7 Figure 2.The two first columns of Figure 2 are the same that those from Figure 1D?

No.

  • Page 7 Section3.2. Line 329. The inhibitor GW4869. Is it water soluble? Have authors employed any solvent (and added to control cells)?

Final DMSO concentration was 0.005% in GW4869 and in the control group (we added comment in section 2.2).

In addition, we followed the reviewer's comments that appeared in the "peer-review" file and corrected to the best of our ability.  The correction are marked in trace changes in the new uploaded file. 

Reviewer 4 Report

Dear Authors,

The study is very comprehensive, detailed and interesting and would be valuable for the field. However, methodology is not properly organized and described. Many methodological details are missing. In the present form, due to the complexity of the whole study, it is difficult to easily understand what and in what order was done. Thus, this section has to be improved and missing information added. Moreover, I have some doubts about the usage of appropriate controls in some parts. Therefore, they have to be explained or described in a more clear way. The same applies to some figures, which do not seem to be properly described and some are even missing. Furthermore, several fragments of the results section are vague to me, so, they need a better description or explanation. Finally, some conclusions are to far reaching in my opinion. Detailed comments are in the pdf file.

Best wishes

Author Response

Thank you so much for the comments for the article "Extracellular Vesicular Transmission of miRNA-423-5p from HepG2 Cells Stimulates De-differentiation of Hepatic Stellate Cells" which was submitted for publication in "cells".

We followed the reviewer's comments that appeared in the "peer-review" file and corrected to the best of our ability.  The correction are marked in trace changes in the new uploaded file. 

Round 2

Reviewer 1 Report

I have no more questions.

Author Response

Thank you so much for the accompaniment in the process of publishing the article "Extracellular Vesicular Transmission of miR-423-5p from HepG2 Cells Inhibits the Differentiation of Hepatic Stellate Cells".

Reviewer 2 Report

All my concerns were cleared.

Author Response

(The authors gave the same response as above.)

Reviewer 4 Report

Dear Authors,

I accept the introduced corrections. The remaining, few minor comments are highlighted in the pdf file.

Best wishes

Author Response

Thank you so much for the accompaniment in the process of publishing the article "Extracellular Vesicular Transmission of miR-423-5p from HepG2 Cells Inhibits the Differentiation of Hepatic Stellate Cells".

All grammar and spelling mistakes ant text clarifications that were highlighted corrected.

R: Are you sure that this is correct? In the previous version you wrote LX2 EVs?

Yes. Thanks for the attention. In this part of the article a comparison was made between different stress conditions using only HepG2-EV (Figure 8).

R: My previous question has not been answered. What was the principle behind the selection of this miRNA - what important lgenes, pathways caught your attention?

The following sentence added to the text: According to the miRBase (the microRNA database) one of the predicted targets for miR-423-5p is Cola1a (Target rank 36). (Line 515)